# How do pilot and feasibility studies inform randomised placebo-controlled trials in surgery? A systematic review

Sian Cousins ,[1] Alexander Gormley ,[2] Katy Chalmers ,[1] Marion K Campbell,[3] David J Beard ,[4] Natalie S Blencowe ,[1] Jane M Blazeby [1]

For numbered affiliations see end of article.

**Correspondence to**
Dr Sian Cousins;
sian.cousins@bristol.ac.uk

## ABSTRACT

**Introduction** Randomised controlled trials (RCTs) with a placebo comparator are considered the gold standard study design when evaluating healthcare interventions. These are challenging to design and deliver in surgery. Guidance recommends pilot and feasibility work to optimise main trial design and conduct; however, the extent to which this occurs in surgery is unknown.

**Method** A systematic review identified randomised placebo-controlled surgical trials. Articles published from database inception to 31 December 2020 were retrieved from Ovid-MEDLINE, Ovid-EMBASE and CENTRAL electronic databases, hand-searching and expert knowledge. Pilot/feasibility work conducted prior to the RCTs was then identified from examining citations and reference lists. Where studies explicitly stated their intent to inform the design and/or conduct of the future main placebo-controlled surgical trial, they were included. Publication type, clinical area, treatment intervention, number of centres, sample size, comparators, aims and text about the invasive placebo intervention were extracted.

**Results** From 131 placebo surgical RCTs included in the systematic review, 47 potentially eligible pilot/feasibility studies were identified. Of these, four were included as true pilot/feasibility work. Three were original articles, one a conference abstract; three were conducted in orthopaedic surgery and one in oral and maxillofacial surgery. All four included pilot RCTs, with an invasive surgical placebo intervention, randomising 9–49 participants in 1 or 2 centres. They explored the acceptability of recruitment and the invasive placebo intervention to patients and trial personnel, and whether blinding was possible. One study examined the characteristics of the proposed invasive placebo intervention using in-depth interviews.

**Conclusion** Published studies reporting feasibility/pilot work undertaken to inform main placebo surgical trials are scarce. In view of the difficulties of undertaking placebo surgical trials, it is recommended that pilot/feasibility studies are conducted, and more are reported to share key findings and optimise the design of main RCTs.

**PROSPERO registration number** CRD42021287371.

## STRENGTHS AND LIMITATIONS OF THIS STUDY

⇒ To our knowledge, this work is original and is the first review to use systematic methods to identify and examine feasibility and pilot work to inform main surgical placebo randomised controlled trials.
⇒ This work has been performed to a high standard; rigorous searches were undertaken and all articles were screened by two reviewers.
⇒ The current review searches are limited to the published literature and it is possible that feasibility and pilot work that may have been conducted but not published.

## INTRODUCTION

Randomised controlled trials (RCTs) with a placebo comparator have the potential to answer key clinical questions regarding the effectiveness of invasive procedures, including surgery. Indeed, exemplar studies have been undertaken that have informed clinical practice.[1 2] Placebo-controlled trials in surgery and other invasive procedures are, however, challenging. There are ethical issues[3 4] related to potential risk to patients, and practical issues regarding the acceptability of the design of the invasive placebo comparator to patients and trial personnel.

Current guidance for the development of complex healthcare interventions[5] and for the design of placebo surgical trials[6] supports feasibility and pilot work to optimise evaluation in a main RCT. Pilot and feasibility work can address uncertainties related to the integrity of the study protocol, recruitment and retention, outcome measures, randomisation procedures, as well as development and acceptability of the intervention itself.[7 8] This is particularly important for placebo surgical trials where acceptability and feasibility may be more challenging. It is unknown to what extent such preparatory work is conducted prior to main surgical placebo trials.

The aim of this review is to examine the extent and type of publications reporting feasibility and pilot work conducted to inform main placebo-controlled trials of surgery, and identify exemplar studies to inform future work.

## METHODS

Published studies reporting feasibility/pilot work conducted in preparation for main randomised placebo-controlled surgical trials were identified through examination of RCTs retrieved by a systematic review. Feasibility/pilot studies identified were then examined in-depth, as detailed below.

### Systematic review of placebo-controlled randomised surgical trials

An existing review[9] which identified placebo surgical trials was updated by extending the searches to 31 December 2020. Searches used the same search terms and electronic databases (Ovid MEDLINE, Ovid EMBASE and CENTRAL)[10] (online supplemental material 1). Additional articles, with no restriction on publication date, were identified by hand searching and expert knowledge.

### Eligibility criteria—placebo-controlled randomised surgical trials

Eligibility criteria are described fully in Cousins *et al* 2019.[9] Briefly, articles reporting randomised trials (including follow-ups, protocols and any self-reported as 'pilot' RCTs) comparing surgery with placebo interventions were included. Surgery was defined as any invasive procedure that changes the anatomy and requires a skin incision or use of endoscopic techniques.[11] Placebo interventions referred to any surgical procedure that was intended to mimic the treatment intervention under evaluation. This included placebo interventions of all types regardless of the degree of invasiveness. Pharmaceutical or dental interventions and reviews were excluded. Protocols of all included studies were retrieved, where available.

### Identifying published pilot/feasibility studies conducted to inform placebo-controlled randomised surgical trials

Main trial publications identified by the systematic review were read fully, including protocols and clinical trials registry entries, where available. Potentially eligible feasibility/pilot studies were identified in two ways. In the first instance, studies referred to explicitly in the main trial text using the terms 'feasibility' or 'pilot', with associated reference(s) were retrieved. Where references were not provided, publication lists of the corresponding author were hand-searched to identify any publications related to work undertaken before the main trial that were cited as being relevant to the feasibility/pilot work mentioned. Second, reference lists of the main trials were hand-searched to identify studies self-reported in the title as 'feasibility' or 'pilot' studies. The use of these terms to identify feasibility/pilot work is supported by a study conducted by Eldridge *et al* 2016 that aimed to develop a conceptual framework for defining feasibility and pilot studies.[7] The study found that of 27 studies identified undertaken in preparation for a RCT, all used at least one of these terms in their titles.[7] Main trial texts were examined by two reviewers (SC and AG) independently. RCTs included in the systematic review update that were self-reported as 'pilot' were also examined for eligibility. Scoping searches on Medline (Ovid SP) electronic database using the search concepts 'feasibility/pilot', 'surgery' and 'placebo', combined with 'and' but with no additional filters, did not retrieve any relevant results. Therefore, additional systematic searches of electronic databases for pilot and feasibility studies were not conducted.

### Eligibility criteria—feasibility and pilot studies to inform main RCTs

For the purposes of this review, eligible studies were those that included an explicit statement that the work was to inform the design and/or conduct of a future main placebo-controlled surgical trial (irrespective of whether the study was labelled or entitled pilot/feasibility or if it had been referred to as pilot/feasibility work in the main trial text). Studies of any design, with aims including, but not limited to, the acceptability of interventions to patients and clinicians, recruitment and retention, and development of invasive placebo interventions, were included. Internal pilot studies (those which solely tested the finalised design as part of the main trial) and those primarily assessing efficacy/effectiveness/safety outcomes were excluded. Original articles and conference abstracts were included; letters, editorials and reviews were excluded. At least two reviewers (SC, AG, KC) screened identified studies independently to ensure they met the above eligibility criteria. Disagreements were resolved by consensus or by a third senior reviewer (JMB), if necessary. Where included feasibility/pilot publications referenced additional related publications, these were retrieved to inform data extraction.

### Data extraction and analysis

Data extracted from included RCTs identified by the systematic review update were: year of publication, region, clinical area (eg, gastrointestinal), number of centres, number of patients randomised and treatment intervention. Data extracted from included feasibility/pilot studies were: publication type (original article or abstract), clinical area and treatment intervention, study design, number of centres, sample size, comparison groups, reported study aims and any text related to the invasive placebo intervention, specifically regarding work done to inform its development and whether studies reported criteria against which decisions would be made to progress to a main RCT. Data were extracted by one reviewer using a standardised data extraction form. A second reviewer extracted data for 20% of articles to identify any potential systematic errors in data extraction. Where multiple articles related to the same study, they were grouped into a single set, and data extraction was conducted on a 'per study' (rather than 'per article') basis.

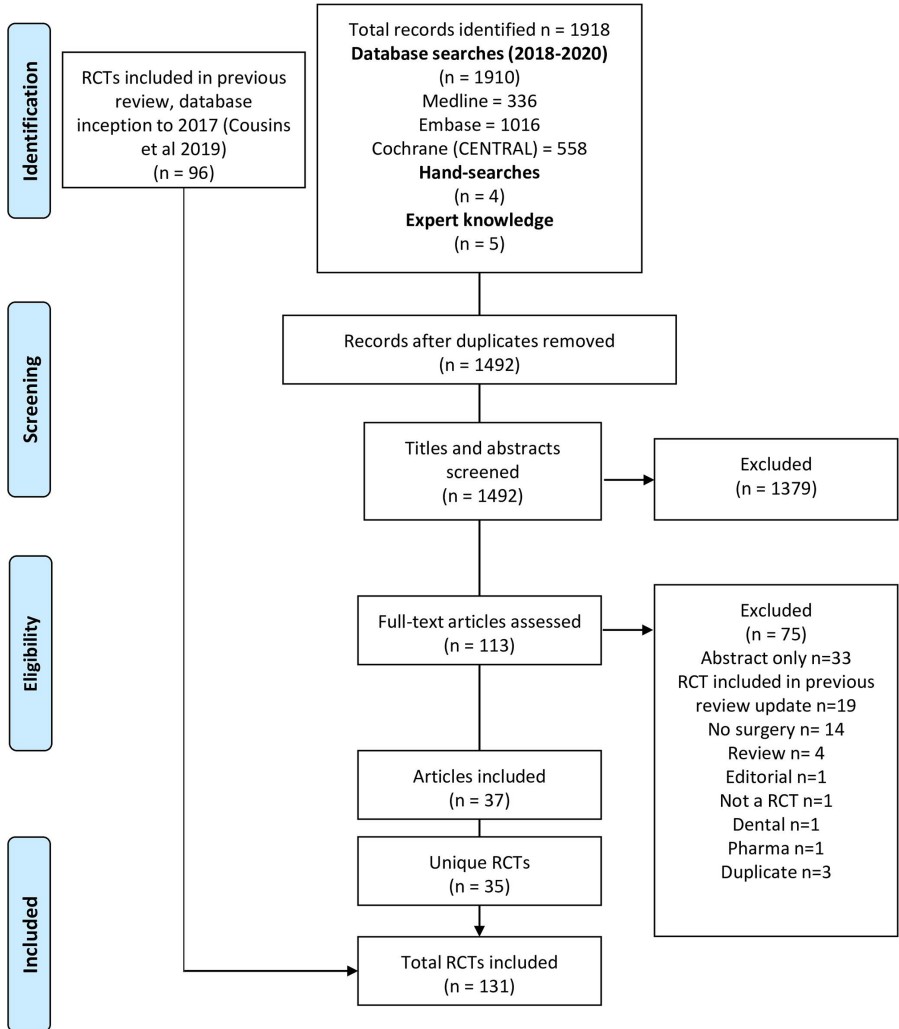

**Figure 1** PRISMA diagram showing screening process of retrieved articles. PRISMA, Preferred Reporting Items for Systematic Reviews and Meta-Analyses; RCTs, randomised controlled trials.

Descriptive statistics summarised basic data where appropriate and verbatim text was summarised descriptively.

### Patient and public involvement
No patients were involved.

### RESULTS
#### Placebo-controlled randomised trials of invasive procedures
The 96 articles identified by the previous review[9] were added to those identified in the current review update. Searches retrieved 1918 articles, of which 113 full texts were screened and 35 trials included (figure 1). Finally included were 131 trials (online supplemental material 2); this included 4 RCTs that were self-reported as 'pilot' identified by the systematic review update. Characteristics are shown in table 1.

#### Feasibility and pilot studies conducted to inform main placebo-controlled surgical RCTs
Of the 127 main placebo RCTs identified (figure 2), 29 referenced studies were explicitly referred to as 'feasibility' or 'pilot' studies in the main trial text. Hand-searching

references of the papers found a further 14 studies self-reported as 'feasibility' or 'pilot' studies in the title. These were combined with the 4 RCTs self-reported as 'pilot' identified in the systematic review (total 47 papers examined). In-depth reading of these found that the majority (n=37) did not report pilot or feasibility work to inform a placebo-controlled main RCT and so were excluded. Although they had been referred to, or were self-reported as, pilot or feasibility studies, there was no explicit statement in the whole report that the work was intended to inform a subsequent main placebo-controlled surgical trial. All of these 37 papers wrote about intentions to examine treatment effects. Two papers reporting 'internal pilot' studies, two duplicates[12 13] (papers that were identified twice, and are included below) and one review[14] and letter[15] were also excluded.

Four studies were explicit in their intention to inform main placebo surgical RCTs design and conduct and were included.[12 13 16 17] All self-reported as a pilot or feasibility study in the title and outlined aims related to their intention to inform a future main placebo-controlled surgical RCT in their introduction. Study details are

**Table 1** Characteristics of placebo-controlled randomised controlled trials of invasive procedures identified in the systematic review update

| Characteristic | Number of RCTs, n=131 (%) |
|---|---|
| Year of publication | |
| ≤2000 | 28 (21) |
| 2001–2010 | 38 (29) |
| 2011–2020 | 65 (50) |
| Region | |
| USA | 44 (34) |
| Mainland Europe | 32 (24) |
| UK | 17 (13) |
| Australia | 9(7) |
| Asia | 7(5) |
| Canada | 3(2) |
| South America | 1(1) |
| Multiregion | 18 (14) |
| Clinical area | |
| Gastrointestinal | 44 (34) |
| Orthopaedics and trauma | 28 (21) |
| Oral and maxillofacial | 22 (17) |
| Interventional cardiology | 10 (8) |
| Cardiothoracic | 7(5) |
| Neurosurgery | 6(5) |
| Gynaecology and obstetrics | 5(4) |
| Ophthalmology | 4(3) |
| Podiatry | 3(2) |
| Urology | 2(2) |
| Number of centres | |
| 1 | 44 (34) |
| 2–5 | 25 (19) |
| 6–10 | 9(7) |
| >10 | 22 (17) |
| Not reported | 31 (24) |
| Number of patients randomised* | |
| 1–100 | 82 (63) |
| 101–200 | 23 (18) |
| >200 | 20 (15) |
| Treatment intervention | |
| Endoscopic | 50 (38) |
| Minimal access | 36 (27) |
| Percutaneous | 31 (24) |
| Open surgery | 14 (11) |

*n=6 included protocols (number of patients randomised not reported).

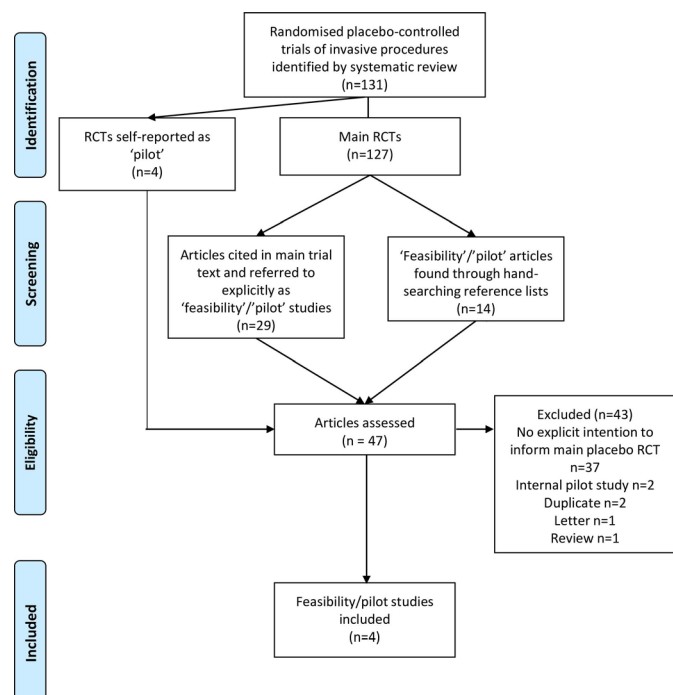

**Figure 2** PRISMA diagram showing screening process of feasibility and pilot studies. PRISMA, Preferred Reporting Items for Systematic Reviews and Meta-Analyses; RCTs, randomised controlled trials.

summarised in table 2 and described individually in detail below.

### Campbell *et al* 2011

This mixed methods study,[17 18] published as an original article and identified by the systematic review update, examined the feasibility of a main trial evaluating arthroscopic lavage compared with placebo surgery and non-operative management for patients with knee osteoarthritis. An initial exploratory phase consisted of in-depth qualitative (interviews and focus groups) (n=257) and quantitative (postal surveys) work (n=780) to explore views and opinions about a main trial. Issues of the design of the invasive placebo comparator, including surgical and anaesthetic components, and the acceptability of this to key stakeholders, including patients, health professionals and chairs of ethics committees, were explored. Broadly, all stakeholders agreed there was a need to investigate arthroscopic lavage further and surgeons and patients expressed uncertainty about the overall effectiveness of the technique. Discussion regarding the design of a placebo intervention centred around which invasive components were required ('The consensus emerged fairly readily that three superficial skin incisions were needed, that these should only pierce the epidermis, and that any penetration of the knee capsule should

**Table 2** Study details of feasibility and pilot studies identified to inform main placebo surgical RCTs

| Author (year) | Publication type | Clinical area (treatment intervention) | Study design (comparator group) | # centres | Sample size | Reported study aims |
|---|---|---|---|---|---|---|
| Campbell (2011) | Original article | Orthopaedics (Arthroscopic lavage for knee osteoarthritis) | Qualitative interviews and focus groups | NA | 257 | Examine the need for a placebo-controlled main RCT; whether an appropriate placebo can be designed (including surgical and anaesthetic components); and the acceptability of an invasive placebo to patients, clinicians and ethics committee chairs |
| | | | Survey | NA | 780 | |
| | | | Pilot RCT (Invasive placebo and no treatment) | 2 | 49 | Examine acceptability of an invasive placebo to health professionals Feasibility of recruitment |
| Kallmes (2002) | Conference abstract | Orthopaedics (Percutaneous vertebroplasty for osteoporotic spinal fractures) | Pilot RCT (Invasive placebo) | NR | 5 | Feasibility of recruitment |
| Powell (2001) | Original article | Oral and maxillofacial (Radiofrequency reduction of turbinate hypertrophy for sleep-disordered breathing) | Pilot RCT (Invasive placebo) | 1 | 22 | Estimate treatment effect and impact of study design Determine appropriate study design of main trial Feasibility of conducting main RCT |
| Moseley (1996) | Original article | Orthopaedics (Arthroscopic debridement for knee osteoarthritis) | Pilot RCT (Invasive placebo) | 1 | 10 | Examine the need for a placebo-controlled main RCT Feasibility of recruitment Develop and test outcome measures Acceptability of placebo Ability of placebo to blind patients |

NA, not applicable; NR, not reported; RCT, randomised controlled trial.

be avoided'), and what form anaesthesia should take. There was general agreement that, assuming general anaesthesia was adopted, inclusion should be limited to low-risk patients. Findings from the survey supported the insights gleaned from discussion in the focus groups. A pilot RCT was also conducted. This randomised 49 patients from 2 centres to examine the feasibility of the proposed placebo-controlled design, including gaining ethical and local approvals and recruitment of patients and delivering placebo surgery. Decisions about whether to progress to a main RCT were reported. The authors concluded that a main placebo surgical trial could be successfully designed and was generally acceptable; however, the considerable barriers faced in conducting the trial in practice meant a main RCT was not feasible and did not take place. These barriers mainly concerned gaining local clinical approval at sites, even after ethical approval was secured. There were concerns regarding indemnity and who would pay for placebo procedures, as well as about the inclusion of surgeons or centres that do not usually offer arthroscopic lavage.

### Kallmes *et al* 2002
This study,[16] published as a conference abstract, examined whether recruitment to a trial comparing vertebroplasty and a placebo for osteoporotic spinal fractures was possible. This was assessed within a pilot RCT that randomised five patients in one centre to treatment or placebo interventions. The placebo intervention included fluoroscopically guided placement of a 25-gauge needle and infiltration of the pedicle with Bupivicaine, without placement of either the vertebroplasty needle or cement, as in the treatment group. Although the methacrylate monomer was opened in the procedure room to give patients olfactory cues to simulate cement preparation. Localised pressure was also placed on the back of patients in the placebo group and operators gave verbal cues typical during cement injection. Although not specified as an explicit aim, the study reported blinding success by asking participants to guess which procedure they had

received; all 5 patients guessed they had received the placebo intervention. The authors concluded that enrolment of patients into a main placebo-controlled trial was feasible.[19]

### Powell *et al* 2001

This single-centre study[12] identified by the systematic review update aimed to assess the impact of inclusion of an invasive placebo comparator on estimates of treatment effect. A pilot RCT randomised 22 participants with sleep disordered breathing to temperature-controlled radiofrequency reduction of turbinate hypertrophy or an invasive placebo. The placebo intervention was identical to treatment, with electrode placement into the anterior inferior turbinate, except that a separate unblinded investigator used a covert radiofrequency energy cut-off switch, to ensure none was applied. The study compared outcomes between blinded and unblinded assessors. It found that unblinded assessment yielded greater treatment effect (bias) and this was used as rationale by the authors for the need to include a placebo comparator in a main trial. Although not explicitly outlined as a study aim authors commented that it was feasible to design a placebo procedure with only the 'active ingredient' withheld. The authors argued that the treatment creates minimal morbidity and it was ethical to conduct a placebo trial because the treatment was not yet the standard of care and there was clinical equipoise. A placebo intervention was also deemed feasible because there were no obvious distinguishable characteristics of treatment, in that its effect was subtle. The authors concluded that a future definitive study was feasible and inclusion of a placebo was critical; however, it is unclear whether it took place as no main trial publication could be found.

### Moseley *et al* 1996

This study,[13] published as an original article, aimed to determine whether a placebo control was necessary in the main RCT, the feasibility of recruitment, the ability of the placebo to blind participants and outcome assessors, and the satisfaction of patients allocated to placebo. They conducted a pilot RCT of 10 patients in 1 centre. The study interventions included arthroscopic debridement of the knee for osteoarthritis, arthroscopic lavage or an invasive placebo procedure. Patients randomised to the placebo received a lesser anaesthetic and did not have an endotracheal tube placed, compared with the two treatment groups that underwent general anaesthetic and placement of an endotracheal tube. The authors reflected that using sedation and local anaesthetic in the placebo group minimised potential complications. Three incisions were made with a scalpel in the placebo intervention, but no instruments were placed in the knee. The knee was, however, manipulated and instruments requested by operators with saline splashed to simulate treatment interventions. Postoperative management was the same across all groups. Surgeons also dictated two operative notes, one for the hospital chart not specifying which

procedure was undertaken, and one kept separately by the surgeon detailing the procedure delivered. Although these measures were taken to maintain blinding, authors did not comment specifically on how the invasive placebo was developed. Patient satisfaction was assessed by asking the questions 'would you recommend the surgery to your friends and family?' and 'do you think the operation was worthwhile?'. Most patients were satisfied in the postoperative period, and at 6 months, seven of the nine patients would recommend the surgery to friends or family. The success of blinding participants and personnel was examined by asking participants and physicians to guess which procedure was performed at all postoperative visits. Patients and outcome assessors were unable to consistently guess which procedure had been delivered. The authors commented that getting study approvals from the necessary committees and institutions was a slow process, with approvals gained only after it was made clear that participants would be fully informed about the placebo-controlled nature of the study. The authors concluded that the main RCT should include an invasive placebo procedure, indeed that failure to use a placebo would 'seriously impair our ability to draw valid inference from the proposed study', and that recruitment was feasible, as was the ability of the placebo to blind patients. A main trial was completed, randomising 180 patients.[1]

## DISCUSSION

Placebo-controlled trials in surgery are challenging to design and deliver and can be contentious. Guidance recommends that pretrial pilot and feasibility work be undertaken to optimise the design and conduct of the trial. Placebo-controlled surgical RCTs identified by the systematic review were rigorously and systematically examined to identify feasibility or pilot work conducted and published to inform the main trial design and conduct. Of the 131 RCTs identified, 47 referred to or referenced 'feasibility' or 'pilot' studies. On detailed scrutiny the vast majority did not state any intention to inform a main placebo-controlled randomised trial, assessing instead the effectiveness/efficacy of the treatment intervention. Four feasibility/pilot studies did outline aims to inform methodological aspects of the main RCT. These focused on key uncertainties of recruitment, the need for a placebo comparator in the main RCT, the ability of the invasive placebo to blind participants and trial persons, the acceptability of an invasive placebo comparator to patients and clinicians, and in one study, the potential components/design of the invasive placebo. All four of the feasibility/pilot studies included a pilot RCT. One[17 18] also employed interviews and focus groups, and a postal survey to examine the design and acceptability of an invasive placebo. The identified studies highlighted the importance of preparatory work and how it can have a major influence on the design of the definitive placebo-surgical trial, both in shaping the final design of the placebo ensuring that it is fit for purpose and able to

blind trial persons effectively and informing the feasibility (or otherwise) of progressing to a main trial. Indeed, one of the studies[18] asserted that a main trial was not feasible; avoiding valuable research resources if a main trial had been conducted that was not successful. The current work found that main placebo-controlled surgical trials are not often informed by published pilot and feasibility studies. It is recommended that feasibility work is conducted and published, not only to optimise the design and conduct of main placebo surgical RCTs, but also to reduce research waste and share lessons learnt.

The limited feasibility and pilot work assessing methodological considerations of placebo surgical RCTs may be due to a historical lack of clarity about the meaning and design of 'pilot' and 'feasibility' studies.[20] Commonly pilot studies have been labelled as such to justify small sample sizes, rather than explore the feasibility of conducting a main RCT. It may also be due to challenges in publishing these studies due to editorial policy; so these studies may have been conducted but not published.[21] Recent work has made strides in clarifying definitions[22 23] and highlights feasibility aims appropriate for assessment in feasibility/pilot studies.[7] The publication of guidance for the reporting of pilot and feasibility studies, including the pilot and feasibility extension of the Consolidated Standards of Reporting Trials,[23–25] and the emergence of journals specific to this area[26] may facilitate the optimisation of the design, reporting and publication of these studies. Guidance for using qualitative research in feasibility studies for trials has also been published.[27] This consists of a list of 16 items within 5 domains (research questions, data collection, analysis, teamworking and reporting) that should be considered when assessing or undertaking qualitative research within feasibility studies for RCTs. Qualitative work has the propensity to add rich information about intervention acceptability and feasibility of delivery; issues paramount when developing and piloting invasive placebo interventions.

Specific to the design and conduct of surgical placebo-controlled trials, published guidelines[6] recommend pilot work to inform the design of invasive placebo procedures, and provides details about how treatment interventions may be deconstructed to identify critical surgical element/s (that can then be omitted from the placebo).[28] The controversial nature of surgical and invasive interventions means that feasibility and pilot work is of the utmost importance. There are issues around the acceptability of the trial to patients, trial personnel and ethics committees due to perceived potential risk. Designing an invasive placebo is also challenging and feasibility work should include examination of which components of the surgery should be included (and omitted) from the invasive placebo intervention and whether it is able to effectively blind participants and trial personnel to trial group allocation.

Inappropriate emphasis on hypothesis testing within pilot and feasibility work, rather than the methodological aspects of the design of a future main RCT is shown in other reviews of published literature. Shanyinde et al[29] identified studies with 'pilot' or 'feasibility' in the title published between 2000 and 2009 and found that of the 50 papers sampled 56% examined methodological issues in-depth and 18% discussed a future trial. Arain et al,[21] who examined published studies found with the keywords 'pilot' and 'feasibility' between 2007 and 2008, found that of the 54 studies identified, 21 reported hypothesis testing and performed statistics to report significant results. Arain et al also searched the United Kingdom Clinical Research Network portfolio for feasibility/pilot studies and of the 34 identified, only 12 tested some component of the research process. A review examining feasibility and pilot studies of surgical interventions funded by the United Kingdom National Institute for Health and Care Research programmes from 2005 to 2015[30] found that although over half of the 35 studies identified examined methodological components, such as recruitment or the current intervention, fewer looked at aspects specific to surgery (n=10). Another review examining published 'pilot' RCTs in orthopaedic surgery found that of the 49 studies included, the majority (n=28) evaluated the efficacy of the intervention.[31]

The current work provides an update to previous systematic reviews of placebo-controlled randomised surgical trials. Further examination of the trials identified by the review update is needed to examine in-depth the methodological aspects, especially as they relate to the ASPIRE (Applying Surgical Placebo in Randomised Evaluations) recommendations,[6] which was published after the previous review.[9] This was outside the scope of this work, but is currently underway. Main trial documents, including protocols and clinical trials registries, were rigorously examined by two independent reviewers to identify feasibility and pilot work. This review did not, however, search for pilot and feasibility studies specifically using electronic databases, although scope searches using search concepts 'feasibility/pilot', 'surgery' and 'placebo' did not identify any additional relevant feasibility/pilot studies. This review was also restricted to published feasibility and pilot studies; it is possible that studies may have been conducted and not published and contacting authors of main placebo controlled trials may be one way to examine this. The exemplar papers identified may be used to inform future feasibility studies in this area. These studies provide useful details about the design and conduct of placebo surgical RCTs specifically, including potential study aims (eg, the ability of the proposed invasive placebo to blind participants and trial persons) and the use of qualitative methods that may inform future RCTs.

There is a dearth of feasibility and pilot work conducted and published to inform the design and conduct of placebo surgical RCTs. Given the challenging nature of these studies, including practical and ethical considerations, feasibility and pilot studies are needed. These will ensure main RCTs are feasible and that the proposed invasive placebo interventions are acceptable and effective in blinding participants and trial persons.

**Author affiliations**
[1]Surgical Innovation theme, Bristol National Institute for Health and Care Research (NIHR) Biomedical Research Centre; Royal College of Surgeons of England (RCSEng) Bristol Surgical Trials Centre, Centre for Surgical Research, Population Health Sciences, Bristol Medical School, University of Bristol, Bristol, UK
[2]Bristol Dental School, University of Bristol, Bristol, UK
[3]Royal College of Surgeons of England, Aberdeen Surgical Trials Centre; Health Services Research Unit, University of Aberdeen, Aberdeen, UK
[4]Nuffield Department of Orthopaedics, Rheumatology and Musculoskeletal Sciences; RCSEng Surgical Intervention Trials Unit; NIHR Oxford Biomedical Research Centre, University of Oxford, Oxford, UK

**Contributors** SC: conceptualisation; methodology; formal analysis; supervision; writing – original draft. AG and KC: formal analysis; writing – review and editing. MKC, DJB and NSB: conceptualisation; methodology; writing – review and editing. JMB: funding acquisition; conceptualisation; metholodology; formal analysis; supervision; writing – review and editing; SC is the guarantor.

**Funding** This work was supported by the National Institute for Health and Care Research Biomedical Research Centre at University Hospitals Bristol and Weston NHS Foundation Trust and the University of Bristol (BRC-1215-20011).

**Competing interests** None declared.

**Patient and public involvement** Patients and/or the public were not involved in the design, or conduct, or reporting, or dissemination plans of this research.

**Patient consent for publication** Not applicable.

**Ethics approval** Not applicable.

**Provenance and peer review** Not commissioned; externally peer reviewed.

**Data availability statement** All data relevant to the study are included in the article or uploaded as supplementary information.

**ORCID iDs**
Sian Cousins http://orcid.org/0000-0003-0088-841X
Alexander Gormley http://orcid.org/0000-0002-2628-9928
Katy Chalmers http://orcid.org/0000-0003-4923-3000
David J Beard http://orcid.org/0000-0001-7884-6389
Natalie S Blencowe http://orcid.org/0000-0002-6111-2175
Jane M Blazeby http://orcid.org/0000-0002-3354-3330

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
