## [Reviewer comments · BMJ Open]

ARTICLE DETAILS

TITLE (PROVISIONAL)	How do pilot and feasibility studies inform randomised placebo-controlled trials in surgery? A systematic review
AUTHORS	Cousins, Sian; Gormley, Alexander; Chalmers, Katy; Campbell, Marion; Beard, David; Blencowe, Natalie; Blazeby, Jane

VERSION 1 – REVIEW

REVIEWER	Hinwood, Madeleine The University of Newcastle, School of Medicine and Public Health
REVIEW RETURNED	27-Jan-2023

GENERAL COMMENTS	Thank you for the opportunity to review this manuscript. Overall it is well written, and placebo trials in surgery is an area which requires more research. However, the overall aim and the study methodology do not completely coincide, and the results are limited. I don't think the full extent of pilot and feasibility work is being captured. Introduction: The aim is broadly "to examine the extent and type of feasibility and pilot work conducted prior to main placebo-controlled trials of surgery"; given that you restricted the search to published feasibility/pilot work only, it is not clear that your study methods answer this question. Much of this pilot work may be unpublished; limiting to published work only will neglect the full extent of pilot work that may inform these study designs. The aim may need to be revised if you only wish to include published pilot work. Methods: Line 106-108: "Placebo interventions referred to any surgical procedure that was intended to mimic the treatment intervention under evaluation". There are often significant differences between studies in terms of how invasive the placebo is, and I think fully defining this is important. Those with more complex placebo designs may require more rigorous pilot testing, for example. If you included all study designs, I think this should still be noted. Line 119: "The use of these terms to identify feasibility/pilot work is supported by a study conducted by Eldridge et al 2018 that aimed to develop a conceptual framework for defining feasibility and pilot studies". Eldridge 2018 was not included in reference list. The reference list seems brief overall. Outcomes were not clearly defined- assume that the primary outcome was to identify firstly which studies included a pilot/feasibility trial, and explore those pilot/feasibility trials
---

	themselves. This should be explicitly stated in a section in the Methods. Results Table 1 isn't referenced; the complete list of studies might be useful to see which these are. When describing the final sample of 4 studies, it might be useful to reference which were the main trial/s that they informed. The description of the sample (starting line 197) is not clear. For example - "All the papers wrote about intentions to examine treatment effects". Here, it is unclear whether you are talking about all the feasibility papers (47 papers) or the 10 which did report pilot or feasibility work; or the 37 which didn't. Unpublished pilot studies- did you consider contacting authors who referred to pilot work but did not publish it? If you are attempting to identify all pilot work, in line with the aim of the study, I think this would be an important step. I imagine a lot of work may not be published. The study methodology fails to identify pilot or feasibility studies that were recently published, but not yet referred to in the main study publication (perhaps because the main study has not yet been conducted, or is currently underway, or because the results of the feasibility study suggested the main study should not go ahead). It depends on the specific research question (is it to link only published studies with their pilot studies? Or to identify all feasibility and pilot work conducted prior to surgical trials [which aligns with the current aim, as it is written; regardless of whether the main trial eventually goes ahead]) Discussion This is reasonable, and provides a good summary of the results.
--	---

REVIEWER	Bernstein, Michael Brown University
REVIEW RETURNED	08-Feb-2023

GENERAL COMMENTS	This was an interesting, well-written manuscript. However, I have the following critiques: 1) Most importantly, I am not convinced that the search strategy of identifying pilot studies was appropriate. The authors seem to only include manuscripts that explicitly stated it was a pilot study for a larger RCT, and n=37 were dropped as a result. I worry that many studies which by all reasonable standards is intended as a pilot would nevertheless not explicitly say so in the manuscript. Similarly, the authors justify their search strategy for "feasibility" or "pilot" in the title from a study by Eldridge which "found that both terms are usually included in the titles and abstracts of studies undertaken in preparation for effectiveness in RCTs." But what does "usually" mean? This could still potentially miss a size-able minority of studies. 2) There was a missed opportunity to provide some more information about the parent RCTs, especially the ones from the updated search.
--

	3) Given that the meat of the manuscript comes down to 4 studies, I would have expected some more detail about them. 4) Minor: I don't see how "The identified studies highlighted the importance of preparatory work" -- why is that?
--	--

VERSION 1 – AUTHOR RESPONSE

Comments from reviewers:

Reviewer: 1

Dr. Madeleine Hinwood, The University of Newcastle Comments to the Author:

Thank you for the opportunity to review this manuscript. Overall it is well written, and placebo trials in surgery is an area which requires more research. However, the overall aim and the study methodology do not completely coincide, and the results are limited. I don't think the full extent of pilot and feasibility work is being captured.

Reply: We thank you for your positive comment. We have addressed your specific comments regarding clarification of the aim and methodological considerations in our replies to the below.

1. Introduction: The aim is broadly "to examine the extent and type of feasibility and pilot work conducted prior to main placebo-controlled trials of surgery"; given that you restricted the search to published feasibility/pilot work only, it is not clear that your study methods answer this question. Much of this pilot work may be unpublished; limiting to published work only will neglect the full extent of pilot work that may inform these study designs. The aim may need to be revised if you only wish to include published pilot work.

Reply: Thank you for this comment. Please find the aim clarified as below. We have clarified that only published work was included. In the discussion section we acknowledge that pilot and feasibility work may have been undertaken but not published (discussion, page 17).

"The aim of this review is to examine the extent and type of publications reporting feasibility and pilot work conducted to inform main placebo-controlled trials of surgery, and identify exemplar studies to advise future work."

2. Methods:

Line 106-108: "Placebo interventions referred to any surgical procedure that was intended to mimic the treatment intervention under evaluation". There are often significant differences between studies in terms of how invasive the placebo is, and I think fully defining this is important. Those with more complex placebo designs may require more rigorous pilot testing, for example. If you included all study designs, I think this should still be noted.

Reply: We agree that invasive placebo interventions may differ widely in how invasive they are. We have updated the methods (page 5) to clarify that placebo designs of all natures were included –

"Placebo interventions referred to any surgical procedure that was intended to mimic the treatment intervention under evaluation. This included placebo interventions of all types regardless of the degree of invasiveness".

3. Line 119: "The use of these terms to identify feasibility/pilot work is supported by a study conducted by Eldridge et al 2018 that aimed to develop a conceptual framework for defining feasibility and pilot studies". Eldridge 2018 was not included in reference list. The reference list seems brief overall.

Reply: Apologies for this oversight. The study by Eldridge was published in 2016 and this reference is included. The reference list has been expanded to include studies reported in the Discussion.

4. Outcomes were not clearly defined- assume that the primary outcome was to identify firstly which studies included a pilot/feasibility trial, and explore those pilot/feasibility trials themselves. This should be explicitly stated in a section in the Methods.

Reply: We apologise for this oversight. We have added the following text to methods, page 4 – “Published studies reporting feasibility/pilot work conducted in preparation for main randomised placebo-controlled surgical trials was identified through examination of RCTs retrieved by a systematic review (PROSPERO registration CRD42021287371). Feasibility/pilot studies identified were then examined in-depth, as detailed below.”

5. Results

Table 1 isn't referenced; the complete list of studies might be useful to see which these are.

Reply: Please find a complete list of the studies identified in the systematic review update included in Supplementary file 2.

6. When describing the final sample of 4 studies, it might be useful to reference which were the main trial/s that they informed.

Reply: Thank you for this comment and the opportunity to clarify. Of the 4 included feasibility/pilot studies, two informed main trials (Kallmes and Moseley) and references for these main trials are included on page 12, line 255, and page 14, line 301, respectively. One (Campbell) found that a main trial was not possible. The remaining study (Powell) concluded that a future definitive study was feasible, however it is unclear whether it took place as no main trial publication could be found (outlined on page 13, lines 270 – 272).

7. The description of the sample (starting line 197) is not clear. For example - "All the papers wrote about intentions to examine treatment effects". Here, it is unclear whether you are talking about all the feasibility papers (47 papers) or the 10 which did report pilot or feasibility work; or the 37 which didn't.

Reply: Thank you for this comment. We have updated the text as below to make this clearer.

Results, pages 8 and 9, “[...] (total 47 papers examined). In-depth reading of these found that the majority (n=37) did not report pilot or feasibility work to inform a placebo-controlled main RCT and so were excluded. Although they had been referred to, or were self-reported as, pilot or feasibility studies, there was no explicit statement in the whole report that the work was intended to inform a subsequent main placebo-controlled surgical trial. All of these 37 papers wrote about intentions to examine treatment effects.”

8. Unpublished pilot studies- did you consider contacting authors who referred to pilot work but did not publish it? If you are attempting to identify all pilot work, in line with the aim of the study, I think this would be an important step. I imagine a lot of work may not be published.

Reply: We agree it would be important, however it was not within the scope of our work. We highlight in the discussion section under methodological limitations that this review was limited to published studies and as such was not able to identify studies not published (Discussion, page 17) – “This review was also restricted to published feasibility and pilot studies; it is possible that studies may have been conducted and not published and contacting authors of main placebo controlled trials may be one way to examine this”

9. The study methodology fails to identify pilot or feasibility studies that were recently published, but not yet referred to in the main study publication (perhaps because the main study has not yet been

conducted, or is currently underway, or because the results of the feasibility study suggested the main study should not go ahead). It depends on the specific research question (is it to link only published studies with their pilot studies? Or to identify all feasibility and pilot work conducted prior to surgical trials [which aligns with the current aim, as it is written; regardless of whether the main trial eventually goes ahead])

Reply: We agree with this sentiment. To be clearer about our work we have clarified the aim of the review as above.

10. Discussion

This is reasonable, and provides a good summary of the results.

Reply: Thank you for this positive comment.

Reviewer: 2

Dr. Michael Bernstein, Brown University

Comments to the Author:

This was an interesting, well-written manuscript. However, I have the following critiques:

1.a. Most importantly, I am not convinced that the search strategy of identifying pilot studies was appropriate. The authors seem to only include manuscripts that explicitly stated it was a pilot study for a larger RCT, and n=37 were dropped as a result. I worry that many studies which by all reasonable standards is intended as a pilot would nevertheless not explicitly say so in the manuscript.

Reply: Thank you for this comment. Our aim was specifically to identify studies that had been undertaken to inform the design of main placebo surgical RCTs, given the methodological challenges of these trials, and so we only included studies that reported this within the manuscripts of the main trials. The aim has been updated to more clearly reflect this (Introduction, page 4, "The aim of this review is to examine the extent and type of publications reporting feasibility and pilot work conducted to inform main placebo-controlled trials of surgery, and identify exemplar studies to advise future work.")

1.b. Similarly, the authors justify their search strategy for "feasibility" or "pilot" in the title from a study by Eldridge which "found that both terms are usually included in the titles and abstracts of studies undertaken in preparation for effectiveness in RCTs." But what does "usually" mean? This could still potentially miss a size-able minority of studies.

Reply: We have now provided more detail regarding the study by Eldridge, which found of 27 studies identified by the authors that were undertaken in preparation for a main RCT, all used at least one of the terms 'pilot' and 'feasibility' in their title (Methods, pages 5 and 6, "The study found that of 27 studies identified undertaken in preparation for a RCT, all used at least one of these terms in their titles..(7)")

2. There was a missed opportunity to provide some more information about the parent RCTs, especially the ones from the updated search.

Reply: We agree with the reviewer. There is further information to be gleaned from examination of the RCTs identified by the review update, however this was outside the scope of this review. Our previous review conducted an in-depth examination of 96 of these trials, with the publication referenced. Work is currently underway to examine other methodological aspects of these trials. We have added text in the discussion to reflect this –

(Discussion, page 17) "Further examination of these trials is needed to examine in-depth the methodological aspects, especially as they relate to the ASPIRE recommendations (6), which was

published after the previous review (9). This was outside the scope of this review but work is currently underway.”

3. Given that the meat of the manuscript comes down to 4 studies, I would have expected some more detail about them.

Reply: Thank you for this helpful comment. We have now added more detail about each of the studies, including further information about the design of each of the placebo interventions and authors reflections on the use of the placebo within a future main trial. Please see Results, pages 11-14.

4. Minor: I don't see how "The identified studies highlighted the importance of preparatory work" -- why is that?

Reply: We have now added further detail about the important aspects of main trial design that feasibility/pilot work can inform (Discussion, page 15), “The identified studies highlighted the importance of preparatory work and how it can have a major influence on the design of the definitive placebo-surgical trial,— both in shaping the final design of the placebo ensuring that it is fit for purpose and able to blind trial persons effectively, and informing the feasibility (or otherwise) of progressing to a main trial. Indeed, one of the studies (18) asserted that a main trial was not feasible; avoiding valuable research resources if a main trial had been conducted that was not successful.”

We hope that you will find the revised manuscript appropriate for publication in BMJ Open

Yours sincerely,
Sian Cousins on behalf of the co-authors

VERSION 2 – REVIEW

REVIEWER	Hinwood, Madeleine The University of Newcastle, School of Medicine and Public Health
REVIEW RETURNED	13-Jun-2023

GENERAL COMMENTS	Thank you for the opportunity to review a revised manuscript. My previous comments have been adequately assessed. The additional information about each of the four studies has improved the manuscript.
--

REVIEWER	Bernstein, Michael Brown University
REVIEW RETURNED	26-May-2023

GENERAL COMMENTS	I thank the author for their careful attention to my prior comments. I remain concerned about the issue of whether there are missing studies, but the added description is helpful. I think the article is otherwise ready for publication, except the following sentence is unclear: "The placebo intervention included fluoroscopically guided placement of a 25-gauge needle and infiltration of the pedicle with Bupivacaine, without placebo of either the vertebroplasty needle or cement, as in the treatment group"
---